# Unlocking the Potential of Stem Cell Microenvironments In Vitro

**DOI:** 10.3390/bioengineering11030289

**Published:** 2024-03-19

**Authors:** Chiara Scodellaro, Raquel R. Pina, Frederico Castelo Ferreira, Paola Sanjuan-Alberte, Tiago G. Fernandes

**Affiliations:** 1Department of Bioengineering and Institute for Bioengineering and Biosciences, Instituto Superior Técnico, Universidade de Lisboa, Av. Rovisco Pais, 1049-001 Lisbon, Portugal; chiara.scodellaro@tecnico.ulisboa.pt (C.S.); raquelramospina@gmail.com (R.R.P.); frederico.ferreira@tecnico.ulisboa.pt (F.C.F.); 2Associate Laboratory i4HB—Institute for Health and Bioeconomy, Instituto Superior Técnico, Universidade de Lisboa, Av. Rovisco Pais, 1049-001 Lisbon, Portugal

**Keywords:** stem cells, cellular niches, microenvironments, biomechanical and biochemical cues, regenerative medicine

## Abstract

The field of regenerative medicine has recently witnessed groundbreaking advancements that hold immense promise for treating a wide range of diseases and injuries. At the forefront of this revolutionary progress are stem cells. Stem cells typically reside in specialized environments in vivo, known as microenvironments or niches, which play critical roles in regulating stem cell behavior and determining their fate. Therefore, understanding the complex microenvironments that surround stem cells is crucial for advancing treatment options in regenerative medicine and tissue engineering applications. Several research articles have made significant contributions to this field by exploring the interactions between stem cells and their surrounding niches, investigating the influence of biomechanical and biochemical cues, and developing innovative strategies for tissue regeneration. This review highlights the key findings and contributions of these studies, shedding light on the diverse applications that may arise from the understanding of stem cell microenvironments, thus harnessing the power of these microenvironments to transform the landscape of medicine and offer new avenues for regenerative therapies.

## 1. Introduction

Stem cells possess a unique ability for prolonged self-renewal and the potential to differentiate into various specialized cell types [1]. These distinctive characteristics make them a highly promising and attractive resource for cell replacement therapies, drug screening applications, and studies in stem cell and developmental biology [2].

Stem cells also hold the potential to support precision medicine approaches for the development more reliable drug disease platforms, without relying on animal models and preparation for tissues for organ replacement. However, the realization of this promise has faced challenges due to the lack of precise control over stem cell fate. Culturing conditions that maintain their unspecialized state or guide them toward specific cell types remain elusive, primarily because our understanding of the endogenous stem cell niches is limited. Complicating matters further, recent discoveries suggest that cells exhibiting stem cell-like properties may play a role in the initiation and sustenance of certain cancers, such as acute leukemia, brain, breast, and skin cancer [3]. Therefore, gaining a deeper understanding of the regulatory mechanisms governing stem cells in their natural niches holds immense potential. This not only expands the repertoire of stem cell-based regenerative medicine treatments, but also opens novel avenues for cancer treatment [4]. For instance, targeting cancer stem cells to direct their destruction could revolutionize strategies against various forms of cancer.

Given the high relevance and complexity of the subject, the stem cell niche currently stands as one of the most crucial and intensely researched areas in the field of stem cell research. However, we are still far from achieving a complete understanding of human morphogenesis, including all the factors and signals present in the niche that influence development. To address this limitation, in vitro models offer a valuable approach to enhance our comprehension of the biological system in both healthy and disease contexts. Various bioengineering methods have been employed to create in vitro models that mimic certain aspects of the human body in terms of structure and function. While these models have achieved a degree of simplification compared to their in vivo counterparts, they still lack essential biomimicry elements such as the intricate interaction between cells and their natural extracellular environment, vascularization, and the spatio-temporal distribution of signals. Presently, multiple research groups are dedicated to developing more realistic in vitro models using diverse bioengineering techniques, including three-dimensional (3D) bioprinting, microfluidic devices, organoids, or combinations of these methods. The discovery of human induced pluripotent stem cells (iPSCs) has significantly propelled progress in this direction.

It is therefore important to acknowledge that stem cell behavior is heavily influenced by the surrounding microenvironment, which consists of a complex network of biochemical, biophysical, and cellular components [5]. By manipulating these microenvironments, scientists can direct stem cells towards specific lineages or activate their regenerative potential [6]. Harnessing the power of biochemical signals, physical cues, and cellular interactions within these microenvironments offers immense potential for addressing previously untreatable conditions and paving the way for transformative therapies that can repair and regenerate damaged tissues and organs. Therefore, knowledge of stem cell microenvironments is crucial for advancing regenerative medicine. It can improve stem cell survival and functionality, facilitate tissue engineering and organ regeneration, aid in disease modeling and drug discovery, enable the regeneration of complex tissues and structures, and open possibilities for personalized therapies. Continued research in this field holds the key to unlocking the full potential of stem cells and revolutionizing the treatment of a wide range of diseases and injuries.

Several research articles have made significant contributions to this field by exploring the interactions between stem cells and their surrounding niches, investigating the influence of biomechanical and biochemical cues, and developing innovative strategies for tissue regeneration. This review focuses on the key findings and contributions of these studies and highlights the emerging technologies for recreating the cellular niche. To make the content more accessible to non-specialists, Table 1 summarizes the most relevant technical terms and abbreviations used in this review.

## 2. The Cellular Microenvironment

Observations from different models have revealed that the stem cell niche regulates stem cell fate through diverse mechanisms (Figure 1). Stem cells are enveloped by the extracellular matrix (ECM), a sugar-rich crosslinked gel network that not only imparts structure and organization, but also delivers biochemical and mechanical cues [7]. Close interactions with supportive cells contribute to short-range signals through soluble factors and membrane proteins [8]. Furthermore, blood vessels closely interface with niches, likely facilitating the transportation of long-range signals and recruiting circulating cells into them [9]. Stem cells within these microenvironments can also respond to inputs from the nervous system, as exemplified in the hematopoietic system [10]. Finally, metabolic signals such as calcium ions or reactive oxygen species (ROS) within the niche can also influence stem cell function [11].

These components collaborate in a complex ensemble either to (a) maintain the stem cell pool in a homeostatic state by promoting asymmetric cell divisions or (b) promote the expansion of the stem cell pool in response to stress or injury through symmetrical self-renewal [12]. Illustrating the significance of this equilibrium, it has been suggested that one potential cause of cancer may be the disruption of precisely orchestrated regulation of cell numbers, leading to the overproduction of stem/progenitor cells through symmetric self-renewal divisions [13]. 

In the following sections, the main components of the niche will be categorized and discussed in these main categories: soluble and immobilized signaling factors, interactions between stem cells and the extracellular Matrix (ECM), direct cell–cell contacts, the physicochemical environment, and biomechanical forces.

### 2.1. Soluble and Immobilized Signaling Factors

Small proteins, such as growth factors, cytokines, and morphogens, play a crucial role as signaling factors that regulate stem cell function in vivo. These molecules exert potent and lasting effects on stem cell fate when presented in a soluble form [14]. The components of the stem cell microenvironment that have been most extensively studied and characterized are soluble molecules and their downstream signal transduction pathways, owing to the ease with which they can be examined. Among the noteworthy soluble molecules are developmental morphogens, including Wingless-INTs (WNTs), hedgehog proteins, fibroblast growth factors (FGFs), and bone morphogenetic proteins (BMPs), which are found in various niches across different species [15]. However, presenting these molecules can be a complex process, particularly when their concentrations are spatially and temporally regulated. An illustrative example of this complexity is the regulation of embryonic development, where distinct signals and molecular pathways sequentially and progressively determine the generation of differentiated cell types [16]. Therefore, understanding these signaling pathways and applying the concept of a stepwise developmental program is crucial when aiming to replicate organ and tissue formation in vitro. 

The secretion of soluble factors by differentiated cells within the niche is a vital contribution to the modulation of stem cell fate. Signals from these differentiated cells can, for example, inhibit the proliferation of stem cells, maintaining a balance among various cell types through negative feedback control—a crucial regulator of stem cell behavior [17]. In vitro, this type of regulation has been observed in cultures of human embryonic stem cells (hESCs). These cells form heterogeneous colonies, not only composed of hESC, but also of hESC-derived fibroblast-like cells contributing to the hESC microenvironment by secreting soluble factors, such as insulin-like growth factor (IGF) II [18]. 

However, while signaling factors have often been integrated into in vitro stem cell culture systems in a soluble form, several studies suggest that the immobilization of these cues is instrumental in mediating their biological function [19]. In vivo, many growth factors and morphogens become immobilized in stem cell niches by binding to the ECM through electrostatic interactions with specific heparin-binding domains [20]. Alternatively, they may bind directly to ECM molecules like collagen or fibronectin [21], or anchor directly to cell membranes, often due to lipid modification of proteins contributing to membrane association [22]. In a general sense, immobilizing these signaling factors alters their local concentration by hindering diffusion and receptor-mediated endocytosis [23], influencing their bioavailability and stability, thus modulating their effects on stem cell fate. A prime example of the crucial role of growth factor immobilization can be found in natural stem cell niches. In neurogenic regions like the subventricular zone, growth factors such as basic FGF are concentrated by heparin sulfate proteoglycan [24], controlling proliferative neural stem cells (NSCs).

Finally, in addition to soluble and immobilized molecules, small metabolite molecules have also been identified as important regulatory cues in stem cell niches [25].

### 2.2. Cell–Extracellular Matrix Interactions

Stem cell function is also regulated by a specialized material secreted by surrounding cells. The so-called ECM serves not only as a structural support, but also as a substrate influencing cell migration, regulating cell morphology, development, and metabolic function [26]. Additionally, it provides signaling cues for self-renewal and differentiation through integrin-mediated activation and downstream signaling events.

The ECM can take the form of a two-dimensional (2D) sheet-like basal lamina or a highly hydrated 3D fibrillar polymer network. It is composed of a complex mixture of various molecules, primarily categorized into two groups: structural proteins and proteoglycans (PGs). The structure and function of any tissue or organ depend on the relative proportion of these two types of constituent molecules [27]. Another crucial set of molecules in this context are integrins, adhesion protein receptors that transmit extracellular signals to stem cells by connecting the niche to the internal cytoskeleton of cells [28].

Key structural proteins within the niche include collagens, elastin, laminin, and fibronectin [29]. Collagens, the most abundant proteins in mammals, play critical roles in cell adhesion and migration during growth, differentiation, morphogenesis, and wound healing. Elastin, being highly insoluble, enables tissues to recover their shape after stretching or contracting. Laminin, a protein facilitating cell adhesion, migration, growth, and differentiation, also plays a significant role. Fibronectin, existing in soluble and insoluble isoforms, binds to integrins and other ECM components like collagen, heparin, fibrin, and PGs. In fact, PGs carry out numerous essential functions in the human body and are considered one of the most critical ECM components for normal cell function and tissue development [30]. Studies with hESCs have reported PGs as among the initial crucial components of ECM during human development [31]. 

PGs consist of a core protein and covalently attached sulphated glycosaminoglycans (GAGs), including chondroitin sulphate and heparin sulphate, as well as hyaluronic acid (HA). HA, a vital component in embryonic development, directly affects tissue organization by interacting with cell surface receptors such as CD44 and the receptor for HA-mediated motility [32]. It is highly expressed in bone marrow stromal cells and on the surface of hematopoietic stem cells (HSCs), directly associated with the regulation of hematopoiesis in the HSC niche [33]. Additionally, while HA activity has been linked to neurogenesis by controlling NSC proliferation and early steps in neuronal differentiation, HA accumulation with aging could impact adult neurogenesis and cognitive functions [34].

### 2.3. Direct Cell–Cell Interactions

Physical contact plays a pivotal role in regulating essential stem cell functions, including anchoring to the niche, modulation of stem cell fate, and mobilization of stem cells to and from the niche [35]. Adhesion often involves cadherins, a family of homophilic adhesion receptors. Additionally, integral membrane proteins like Ephrin and Notch receptors, along with their respective ligands, contribute to cell-contact-mediated signaling between stem cells and their microenvironment. An illustrative instance highlighting the significance of cell–cell contacts in stem cell niches is the adhesive attachment of HSCs to osteoblasts through E-cadherin-mediated interactions [36]. This not only provides anchorage but also serves as a critical component of the HSC niche.

Juxtacrine activation of Notch signaling, facilitated by cell-presented ligands Jagged or Delta, has been implicated in various stem cell niches, including the HSC niche [37]. Likewise, Ephrin-mediated cellular contact between NSCs and neighboring cells is suggested to modulate signaling involved in neurogenesis and NSC self-renewal in the adult brain [38]. Moreover, recent studies have revealed significant interactions between the endothelial cells of the vasculature and embryonic and adult NSCs. The former are responsible for enhancing NSC proliferation through the increased generation of junctional contacts between NSCs [39]. These findings underscore the critical role of cellular organization and density in the cellular niche.

### 2.4. Physicochemical Environment

The physicochemical milieu, encompassing factors such as oxygen tension and pH, constitutes a pivotal element within the cellular microenvironment, exerting significant influence over the regulation of stem cell fate and viability [40]. 

Precisely determining the in vivo oxygen tension of a given tissue presents a formidable challenge [41]. Nonetheless, it is well established that various adult tissues encounter oxygen tensions markedly lower than those prevalent in the surrounding ambient air. Consequently, a hypothesis has emerged postulating that the low oxygen tensions characterizing stem cell niches confer a selective advantage conducive to their biological functions [42,43]. In particular, cells engaged in aerobic metabolism confront oxidative stress due to the generation of ROS, which, in turn, can inflict damage upon DNA [44]. The strategic homing of stem cells within niches characterized by low oxygen tensions potentially shields them from such detrimental effects. Significantly, hypoxia, or low oxygen levels, has been observed to instigate molecular mechanisms serving as crucial regulators in diverse stem cell systems [45].

In fact, cell responses to low oxygen levels are regulated by HIF proteins [46]. Particularly, HIF-1 is a heterodimeric transcription factor consisting of a constitutively expressed β-subunit and an oxygen-regulated α-subunit. Under low oxygen tensions, HIF-1α is stabilized, forming dimers with HIF-1β. This complex translocates to the nucleus, assuming the role of transcriptional activators for an assorted array of genes, some of which hold paramount importance in sustaining the stem cell pool [47].

### 2.5. Mechanical Forces

In addition to the conventional considerations of regulatory effects originating from soluble, cellular, and ECM factors on stem cell fate, contemporary research has significantly broadened the landscape of this domain by integrating the nuanced biophysical characteristics of the microenvironment [48]. Mechanical forces are now recognized as a pivotal regulatory determinant within the stem cell niche, where the deliberate application of mechanical strain has demonstrated effects on both stem cell self-renewal and differentiation [49]. Crucial among the biomechanical features defining this microenvironment are stiffness, shear force, and cyclic strain.

The stiffness of a given material finds common quantification through the apparent Young’s modulus. The inherent diversity in the stiffness of organs and tissues arises from differences in ECM composition, cross-linking density, and mineralization [50]. Furthermore, cell membranes exhibit varying stiffness depending on cell type and differentiation stage. One example is the two-fold greater stiffness of mesenchymal stromal cell (MSC) membranes compared to osteoblasts [51]. Within cells, structures like integrins and focal adhesions orchestrate mechanotransduction processes that decisively shape cell morphology, migration, proliferation, and differentiation [52].

Shear stress also emerges as a critical mediator for an array of vascular and circulating cells [53], encompassing endothelial cells, smooth muscle cells, and leukocytes, for example. Consequently, stem cells strategically situated in close proximity to the vasculature, exemplified by MSCs, become subjects of regulatory influence exerted by shear stress [54]. Interestingly, in vitro exposure of MSCs to shear stress augments essential processes underpinning vasculature formation, including proliferation, endothelial differentiation, and the production of angiogenic factors [55]. Adding another layer to the complexity, cells adjacent to vasculature also grapple with cyclic strain or repetitive stretch, a consequence of pulsatile blood flow [56].

Integral to the definition of the cellular microenvironment is its architectural framework and spatial organization [57]. The niche architecture, marked by nuances in geometry, topography, and dimensionality, adds another layer of complexity. Natural stem cell niches present distinctive geometries sculpted by the spatial distribution of neighboring cells [58]. Notable is the intricate geometry characterizing the sub-ventricular zone and ventricular zone of the mammalian brain [59], where NSCs find themselves ensconced by ependymal cells. Lastly, beyond geometric cues, cells in their native niches encounter diverse topographies, ranging from fibrous ECM to the rugged landscape of mineralized bone, each contributing to the multifaceted orchestration of cell behavior [60].

## 3. Recent Advances in In Vitro Microenvironment Modeling

In the ever-evolving landscape of stem cell research, traditional 2D cell culture systems have long been the stalwart tools for investigating cell behavior. However, their limitations in replicating the intricate physiological cellular organization and biophysical properties found in vivo have led researchers to explore more dynamic avenues. Particularly, 3D models have been developed to deepen our understanding of stem cell microenvironments [61]. These models aim to faithfully recreate the 3D microenvironments in which cells thrive naturally, specifically the macro-, micro-, and nanoscale features of the niche influencing cell responses (Figure 2 and Figure 3).

### 3.1. Emerging Technologies for Recreating the Cellular Niche

The cellular response is highly determined by the characteristics of the scaffold in which the cells reside. Matrix mechanics, degradability, microstructure, and the presence of cell-adhesive ligands become the fundamental qualities shaping the destiny of resident cells [64]. Moreover, the dynamic interplay with other cell types and cell–cell interactions add to the complexity of signals orchestrating the fate of cells in the niche. 

For example, Anthon et al. describe in detail the techniques that can be used to simulate the native tissue microenvironment. From manual assembly techniques crafting porous scaffolds, to cutting-edge technologies like bioprinting and electrospinning, the possibilities are as diverse as the cellular ecosystems they aim to mimic [62] (Figure 4). Particularly, the use of microfabrication techniques in stem cell biology and tissue engineering was highlighted in a comprehensive review article [63]. The authors describe the fabrication methods, biomaterial choices, and applications of microengineered platforms for stem cell culture, differentiation, and tissue development. They discuss how the integration of microscale technologies with stem cell biology offers unprecedented control over cell fate and tissue morphogenesis. This review article provides valuable insights into the potential of microfabrication techniques for advancing regenerative medicine. In one such case, Ramos-Rodriguez et al., for instance, showcase a system built on electrospun polycaprolactone fibers laden with bioactive compounds, perfectly replicating the intricate microenvironment of the skin [65]. These constructs could deliver key bioactive compounds that can enhance skin regeneration and ultimately aid in the development of a complex wound-healing device.

Additive biofabrication also holds great potential for recapitulating the complexity and heterogeneity of tissues and organs. Such methods are crucial for the creation of in vitro 3D models and development of regenerative medicine applications [66]. Specifically, the use of 3D bioprinting techniques allows one to precisely control the shape and composition of a manufactured construction. In here, the proper development of bioinks able to recapitulate the cell microenvironment is crucial, while promoting accurate printability, fidelity and cell viability and function. Moreover, it allows the addition of cells or soluble growth factors to the manufacturing material or bioink, since fabrication typically takes place using mild conditions. Therefore, the choice of materials is critical, and a suitable bioink should have some essential requirements, including printability, biocompatibility, biomimicry, and adequate rheological, chemical, and mechanical properties [67]. Hydrogels derived from natural polymers like cellulose, chitosan, alginate, and collagen, can emulate the natural ECM, providing a nurturing environment for a diverse array of cell functions [68]. Therefore, the combination of 3D bioprinting techniques including inkjet, extrusion and laser-assisted bioprinting, with careful selection of bioinks, allows for precision in shaping and creating these manufactured structures.

Laser-assisted bioprinting was used by Gruene et al. to recreate adipose tissue in vitro [69]. Human adipose-derived stem cells were used, and no detrimental effect of printing was reported on the cell growth. In this work, the authors achieved a multi-cellular graft mirroring the in vivo stem cell niche. 

3D printing and bioprinting also allow one to generate more complex structures. For example, Philippi et al. reported the recreation of bone structures using this technique [70]. Braham et al. even ventured into replicating the bone marrow microenvironment using the 3D printing of pasty calcium phosphate cement and seeded MSCs [71]. The resulting constructs emulate the endosteal niche while ECM loaded with endothelial and MSCs was used to mimic the perivascular niche. The intestinal epithelium topography was also reproduced using high-resolution stereolithography 3D printing [72]. Creff et al. were able to faithfully recreate villi-like structures reminiscent of the crypts of Lieberkühn, the natural invagination where intestinal stem cells reside, and contribute to the constant renewal of the intestinal mucosa.

Still, amidst these breakthroughs, the delicate balance of developing bioink formulations that satisfy both biological and physicochemical requirements remains elusive [73]. Cell damage and loss of cell function can occur due to the mechanical stress caused by the printer during the deposition and exposure of the cell-encapsulated material to chemical crosslinkers for extended periods of time. It is therefore important to carefully control the flow and nozzle speed, the gelation method, and the printing temperature [73].

**Figure 4 bioengineering-11-00289-f004:**
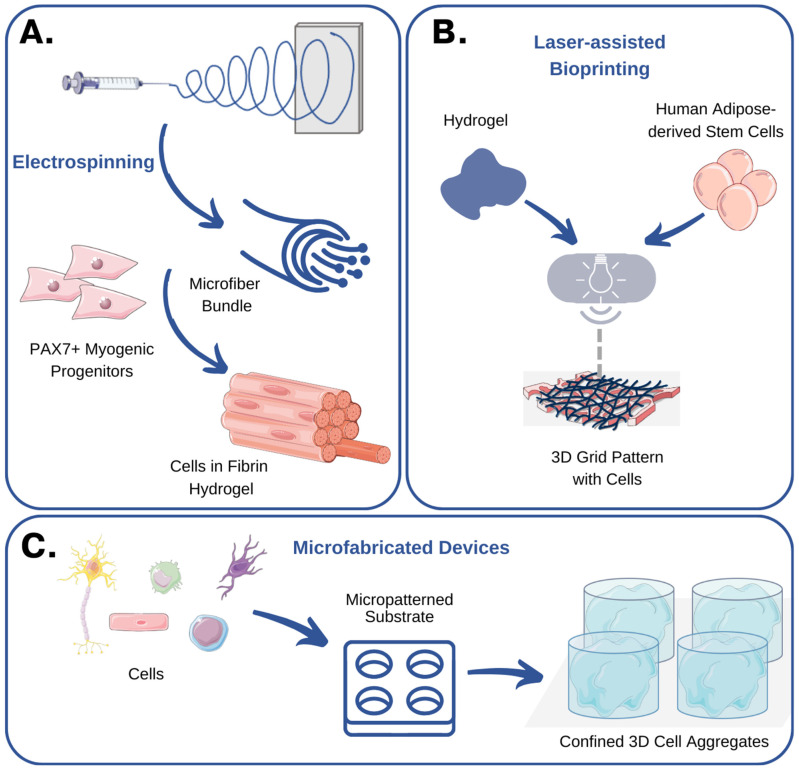
Emerging technologies for recreating the cellular niche: examples and case studies. (**A**) Somers et al. used electrospinning to produce fibrin microfiber bundles that could be populated with myogenic progenitors. These 3D skeletal muscle grafts exhibited myotube formation and expression of muscle-specific markers [74]. (**B**) Gruene et al. used laser-assisted bioprinting to recreate adipose tissue in vitro [69]. No detrimental effect of printing was reported on cell viability and growth. (**C**) In a seminal study, Gottwald et al. used microfabricated devices to confine cells inside microwells and produce 3D cell aggregates [75]. The figure was created using Servier Medical Art, provided by Servier, licensed under a Creative Commons Attribution 3.0 unported license.

### 3.2. Examples and Case Studies

When trying to look for solutions in the field of bone tissue engineering, researchers utilized poly(glycerol sebacate) elastomer to fabricate biocompatible scaffolds with tailored mechanical properties [76]. The incorporation of decellularized bone ECM enhanced the osteoinductive potential of the scaffolds, promoting the osteogenic lineage commitment of MSCs. By controlling the pore size and composition of the scaffolds, researchers achieved improved cell attachment, osteogenesis, and mechanical strength. These findings pave the way for the development of advanced bone tissue engineering strategies that can enhance bone regeneration and repair. In another study focused on the engineering of 3D skeletal muscle grafts using myogenic progenitors and advanced biomaterials, the authors combined small molecules and electrospun fibrin microfiber bundles to successfully generate functional skeletal muscle grafts that exhibited myotube formation and expression of muscle-specific markers [74]. In vivo experiments showed promising results in terms of promoting muscle regeneration. These findings demonstrate the potential of combining myogenic progenitors and biomaterials for engineering functional skeletal muscle grafts and advancing the field of muscle tissue engineering. In a similar fashion, Baumgartner et al. took advantage of coaxial electrospinning to generate meshes of fibers with different orientations and replicate the mechanical properties of tendons [77]. Human adipose-derived stem cells were seeded on the fibers and subjected to various stretching conditions. This study demonstrated that the elastic modulus of the cell-seeded meshes increased over time, particularly in the random fiber group. Random fibers also exhibited a higher level of tenogenic commitment compared to aligned fibers. Stretching resulted in increased expression of pro-inflammatory markers, and cells cultured on random meshes showed significant upregulation of genes associated with tenocyte differentiation.

In another example, Koyanagi et al. explored the interactions between HSCs and MSCs in the bone marrow microenvironment [78]. The study focused on specific genetic mutations in MSCs that can exacerbate hematopoietic neoplasms. By using a decellularized bone scaffold and a clustered regularly interspaced short palindromic repeats-associated protein 9 (CRISPR-Cas9) activation library, the researchers identified candidate factors and successfully engrafted MSCs into the decellularized biomaterials. This work opens possibilities for creating models of the bone marrow niche while deepening our understanding of the interactions between HSCs and MSCs in hematopoiesis and disease development.

Pluripotent stem cells (PSCs) are also essential for future applications in regenerative medicine. However, several challenges persist that affect the translation to more practical solutions, particularly the capacity to control fluctuations in the outcome of the production of PSCs and their derivatives [79,80]. To help summarize these questions, one review focused on the effects of culture-induced fluctuations in the outcome of PSC quality [81]. By understanding the mechanistic basis of how PSC behaviors are altered in response to biomechanical microenvironments, researchers could optimize the bioprocessing of PSCs and their derivatives. For example, chronic kidney disease poses significant societal challenges and PSCs offer a potential solution by enabling the engineering of kidney tissues in vitro. Synthetic and natural polymers have been used to create fibers that promote cell interactions specific to the native environment of the kidney [82]. Combining electrospinning with bioprinting could also lead to the development of more organized, mature, and reproducible kidney organoids. 

Finally, the corneal epithelial stem cell niche and its role in corneal regeneration is another example of a system that has received much attention [83]. Epithelial stem cells residing in this specialized niche are crucial for maintaining the health of the corneal epithelium. Therefore, understanding the characteristics and behavior of these cells has significant implications for corneal regeneration strategies, especially in cases of corneal trauma and diseases.

## 4. Future Directions

The current landscape of 3D bioprinting faces significant challenges in the quest to engineer fully functional tissues and organs. A crucial obstacle lies in formulating bioinks with the requisite characteristics for effective tissue engineering, as highlighted by Raees in 2023 [84]. The difficulties primarily stem from the mechanical weakness of natural biomaterials even after cross-linking, the limitations of synthetic polymers in terms of biocompatibility and cytotoxicity, and the inability of any single component to possess all the necessary properties to replicate the native functions of the ECM.

In overcoming these hurdles, researchers are exploring composite bioinks, as demonstrated by Liu et al. in 2020, incorporating diverse small molecules, growth factors, and nanoparticles into the matrix [85]. This approach aims to enhance the printed structures for specific purposes and prevent composite bioinks from succumbing to shear stress during extrusion, while still promoting cell growth, as proposed by Raees et al. [84]. Similarly, advancements in the miniaturization of cell-based models have the potential to revolutionize this field, while reducing the consumption of cells and reagents. For example, work by Jongpaiboonkit emphasizes the need for future development to focus on automated, high-throughput methods for studying cellular microenvironments and growth conditions in 3D [86]. Array-based formats, as demonstrated by Gottwald et al. [75], and Liu and Roy [87], offer enhanced-throughput platforms for 3D cell culture, reflecting efforts to better emulate in vivo functions.

Crucial to these developments is the understanding of cell–cell and cell–ECM interactions, acknowledging their pivotal role in controlling cell behavior. The incorporation of biosensing elements, exemplified by the work of Jones et al. in 2008, enables local detection of secreted cellular products [88]. These innovations hold promise for creating more in vivo-like structures and improving the sensitivity of detection methods, potentially advancing cell-based drug discovery and target validation. Consequently, the reliability of in vitro assays in predicting in vivo responses is expected to increase, fostering greater adoption and significance of these models in biomedical experimentation [89].

Looking ahead, the field anticipates overcoming existing handicaps by refining bioink formulations, enhancing composite bioinks, and leveraging automated, high-throughput methods for studying cellular microenvironments. These efforts are likely to pave the way for more sophisticated 3D bioprinting and the production of artificial microenvironments in vitro, with the potential to transform the landscape of tissue engineering and drug discovery [90,91].

## 5. Conclusions

Understanding and harnessing the power of stem cell microenvironments has the potential to transform the landscape of medicine and offer new avenues for regenerative therapies. By unraveling the intricate interactions between stem cells and their surrounding niches, researchers are paving the way for the development of targeted interventions and personalized treatments. The contributions of these studies discussed above highlight the importance of investigating stem cell microenvironments and provide valuable insights into the development of regenerative medicine strategies [92]. With continued advancements in this field, we can look forward to a future where stem cell-based therapies become an integral part of medical practice, improving the lives of countless individuals.

## Figures and Tables

**Figure 1 bioengineering-11-00289-f001:**
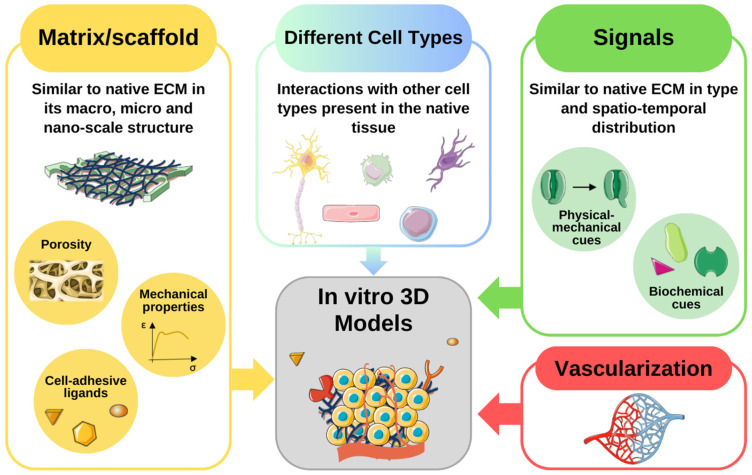
Fundamental elements to include in an 3D in vitro model. The scaffold should be similar to the native extracellular matrix (ECM) in its macro-, micro- and nano characteristics. Some aspects to consider in the creation of the artificial matrix are its porosity, its mechanical properties, and the presence of cell-adhesive ligands. The niches are usually composed of different cell types and this feature should be reproduced in the artificial model. There should be biochemical and physical-mechanical signals like those present in the native tissue. Furthermore, their spatio-temporal distribution is a crucial factor in determining the fate of the cells present in the niche. Finally, in a 3D system, the presence of vasculature is essential to supply nutrients and oxygen, and eliminate waste products. The figure was created using Servier Medical Art, provided by Servier, licensed under a Creative Commons Attribution 3.0 unported license.

**Figure 2 bioengineering-11-00289-f002:**
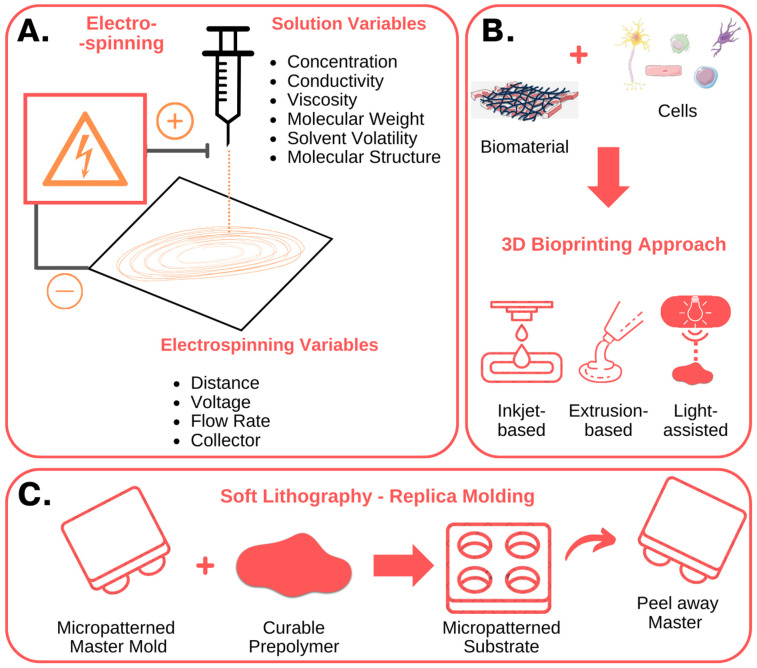
Nanofabrication techniques used to create in vitro stem cell niches: fundamental characteristics. (**A**) Electrospinning involves the use of an electrical potential to produce nano- and micro-meter scale fibers from polymer solutions. By modifying the solution, process and environmental parameters, it is possible to control the diameter and orientation of the fibers, obtaining structures with characteristics like the native ECM. (**B**) Three-dimensional bioprinting techniques, including inkjet-based, extrusion-based, and light-assisted 3D printing, enable the fabrication of complex 3D structures, composed of multiple bioinks, cell types and biomolecules. (**C**) Soft lithography, or replica molding, uses a micropatterned master to mold and shape a polymeric film. The polymer can be curated to produce and fabricate micropatterned substrates with micrometer features, like microwell arrays or microfluidic devices. The figure was created using Servier Medical Art, provided by Servier, licensed under a Creative Commons Attribution 3.0 unported license.

**Figure 3 bioengineering-11-00289-f003:**
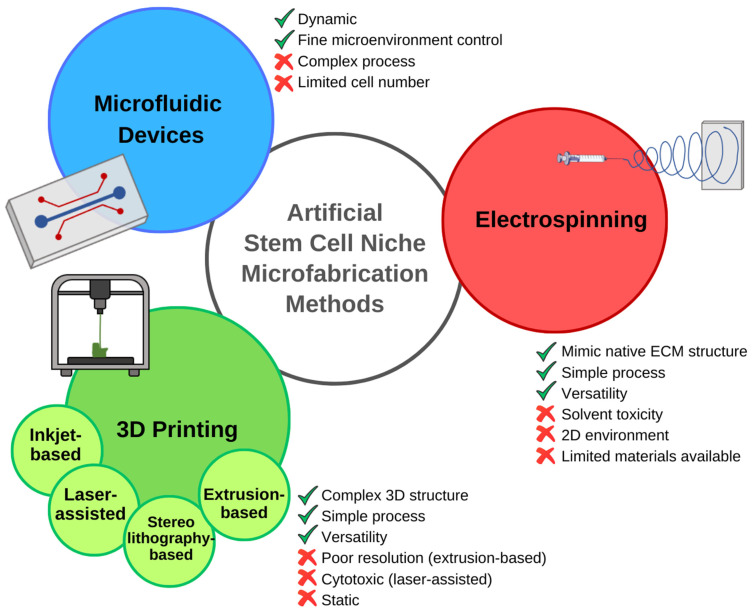
Nanofabrication techniques used to create in vitro stem cell niches: advantages, and disadvantages. Electrospinning is a relatively simple and versatile process that can mimic certain aspects of the native ECM. The toxicity of the solvent, and the high solubility and low molecular weight of the polymer required for the process, limit the use of natural polymers. Moreover, the generated scaffold, although composed of multiple layers, cannot be considered a 3D structure. Three-dimensional printing techniques, including inkjet-based, laser-assisted, stereolithography-based, and extrusion-based 3D printing, enable the fabrication of complex 3D structures. The limitations of this manufacturing method depend on the chosen technique. Extrusion-based 3D printing is characterized by a very simple process, counterbalanced by poor printing resolution. The high precision achieved with laser-assisted techniques goes along with other disadvantages, such as cytotoxicity. Microfabricated microfluidic devices overcome one of the limitations of the other systems, recreating a dynamic microenvironment that allows accurate reproduction of spatio-temporal signaling distribution. However, the complex fabrication process and the restricted size and cell numbers limit these systems, causing scientists to prefer other techniques for several applications [62,63].

**Table 1 bioengineering-11-00289-t001:** Glossary of technical terms and abbreviations.

Technical Terms and Abbreviations	
2D	Two-dimensional
3D	Three-dimensional
Biocompatibility	The ability to interact with a living system without producing an adverse effect
Bioink	A specialized material used in 3D bioprinting, which serves as the medium through which cells are deposited layer by layer to build complex tissue structures
Biomaterial	A natural or synthetic substance that interact with biological systems
Bioprinting	An advanced technology that enables the fabrication of 3D biological structures using living cells, biomaterials, and bioactive molecules. The process involves layer-by-layer deposition of materials capable of incorporating living components to create tissues, organs, and other biological constructs. Specific methods include inkjet, extrusion, and light-assisted bioprinting
BMPs	Bone morphogenetic proteins, a group of signaling molecules part of the transforming growth factor-beta (TGF-β) superfamily
Cell niche	Refers to the specialized microenvironment in which cells reside within tissues or organs. It encompasses the physical, chemical, and biological factors that regulate the behavior, maintenance, and fate of cells
CRISPR-Cas9	Clustered regularly interspaced short palindromic repeats and CRISPR-associated protein 9, a gene-editing technology that allows to make precise changes to an organism’s DNA
Cytotoxicity	The capacity of an agent to cause damage or death to cells
Differentiation	The process by which an immature cell becomes specialized, acquiring specific structures and functions that enable it to perform particular tasks within an organism
ECM	Extracellular matrix, a complex 3D network of proteins, glycoproteins, proteoglycans, and polysaccharides that surrounds and supports cells within tissues and organs in multicellular organisms
Electrospinning	A technique capable to produce ultrafine fibers. The process involves the use of an electric field to draw charged polymer solutions into thin fibers that are collected on a grounded substrate
FGFs	Fibroblast growth factors, a family of signaling proteins that bind to specific cell surface receptors and are involved in various biological processes, including cell growth, proliferation, differentiation, and tissue repair
GAGs	Glycosaminoglycans, a family of polysaccharides that are major components of the ECM. Examples are hyaluronic acid (HA), chondroitin sulfate, and heparan sulfate
Hedgehog	A family of secreted signaling proteins that play essential roles in embryonic development, tissue homeostasis, and stem cell regulation across various species, including humans
hESCs	Human embryonic stem cells, pluripotent stem cells derived from the inner cell mass of the blastocyst, an early stage of embryonic development
HIFs	Hypoxia-inducible factors, a family of transcription factors that regulate the cellular response to changes in oxygen levels
High-throughput	The capability of performing many analyses in parallel, typically using automation, miniaturization, and advanced technologies
HSCs	Hematopoietic stem cells, multipotent stem cells that give rise to all types of blood cells in the body
Hydrogel	A 3D network of hydrophilic polymer chains that are capable of absorbing and retaining large amounts of water
Hypoxia	Low oxygen levels
IGF	Insulin-like growth factor, peptide hormone with structural similarities to insulin. Plays essential roles in regulating growth, development, metabolism, and cellular function in various tissues throughout the body
iPSCs	Induced pluripotent stem cells, a type of pluripotent stem cell that can be generated from somatic cells through a process of cellular reprogramming
Microfluidic devices	Miniaturized platforms that manipulate small volumes of fluids at the microscale level
Microwell arrays	Microscale platforms composed of arrays of small wells or compartments arranged in a regular pattern on a substrate. Commonly fabricated using microfabrication techniques such as soft lithography or replica molding
MSCs	Mesenchymal stem/stromal cells, multipotent cells that can differentiate into a variety of cell types, including bone, cartilage, fat, and other connective tissue cells
NSCs	Neural stem cells, a type of stem cell found in the nervous system. Can differentiate into neurons, astrocytes, and oligodendrocytes
Organoids	3D miniature organ-like structures that are derived from stem or progenitor cells and exhibit rudimentary organ function and organization
PGs	Proteoglycans, a type of glycoprotein found in the ECM of tissues. They consist of a protein core to which GAGs are attached
Printability	The feasibility of a given material to be use in a printing process
PSCs	Pluripotent stem cells, a type of stem cell that can differentiate into all cell types in the body. Include ESCs and iPSCs
Regenerative medicine	A multidisciplinary field that aims to restore, repair, or replace damaged tissue or organs in the body
ROS	Reactive oxygen species, are chemically reactive molecules containing oxygen
Scaffold	3D structure or framework that provides mechanical support, guidance, and a conducive environment for cells to attach, grow, and differentiate
Shear stress	A mechanical force exerted parallel to the surface of an object or fluid layer
Soft lithography	A set of techniques used in microfabrication to pattern and fabricate structures on the micrometer scale using elastomeric materials as stamps or molds
Soluble factors	Molecules or compounds that are soluble in biological fluids, and play critical roles in cellular signaling, communication, and regulation of physiological processes
Stem cell	Undifferentiated cells with the capacity to self-renew and ability to differentiate into various specialized cell types
Stiffness	A property that refers to the resistance of a material to deformation in response to an applied force or load
WNTs	Wingless INTs, a family of highly conserved signaling molecules that play crucial roles in embryonic development, tissue homeostasis, and adult stem cell regulation

## Data Availability

Not applicable.

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
