# Peer review of "Unlocking the Potential of Stem Cell Microenvironments In Vitro"

_bioengineering, 2024, doi:10.3390/bioengineering11030289_

Round 1

Reviewer 1 Report

Comments and Suggestions for Authors

In this manuscript, “Unlocking the Potential of Stem Cell Microenvironments in Vitro” by Scodellaro et al. highlights the key findings and contributions of these studies, shedding light on the diverse applications that may arise from the understanding of stem cell microenvironments. Although authors had collected many literatures, it still not easy to understand the importance of this topic without some more tables or figures. Therefore, I would suggest that authors may take a major revision. Here are the comments and suggestions:

1.         In Fig. 1, authors are suggested to adapt some more figures from literatures.

2.         In Fig. 2, please list the details of each techniques from literature.

3.         In the section of 3.2 Examples and Case Studies, some more figures can be adapted from literatures.

Author Response

In this manuscript, “Unlocking the Potential of Stem Cell Microenvironments in Vitro” by Scodellaro et al. highlights the key findings and contributions of these studies, shedding light on the diverse applications that may arise from the understanding of stem cell microenvironments. Although authors had collected many literatures, it still not easy to understand the importance of this topic without some more tables or figures. Therefore, I would suggest that authors may take a major revision. Here are the comments and suggestions:

  1. In Fig. 1, authors are suggested to adapt some more figures from literatures.

  1. In Fig. 2, please list the details of each technique from literature.

  1. In the section of 3.2 Examples and Case Studies, some more figures can be adapted from literatures.

RE:

We acknowledge the comments of the reviewer.

To address these issues, we have revised the figures.

Figure 1 was revised for quality and clarity.

We have included a new Figure 2 with schematics of the different engineering approaches for cell niche production in vitro.

Figure 3 is now summarizing pros and cons of each technique mentioned in the text.

Finally, we have produced a new figure, Figure 4, which highlights a few examples referenced in the main text.

Reviewer 2 Report

Comments and Suggestions for Authors

The review focuses on the important role of stem cell microenvironments in regenerative medicine, highlighting advancements in understanding and manipulating these environments for improved treatment strategies. It summarizes the role of the extracellular matrix, biochemical signals, and mechanical cues in influencing stem cell differentiation and function. In addition, this review also discusses innovative techniques such as 3D bioprinting and organ-on-a-chip models that mimic these complex microenvironments for enabling better understanding and control over tissue development and regeneration. The review also covers a bit of detail regarding the potential of these technologies in developing personalized medical treatments and their challenges, including ethical considerations and the need for further research to optimize stem cell-based therapies. Overall, this review underscores the importance of integrating multidisciplinary approaches to unlock the full potential of stem cell research in addressing current limitations in regenerative medicine. I think this review is worthy to be further considered. The authors shall better organize the content by tables/figures and include more relevant articles in the discussion. Please see my comments below.

1.      It will be helpful to include a glossary for footnotes to explain technical terms and abbreviations to make the content more accessible to non-specialists.

2.      Please provide a table explaining the pros/cons/limitations of different engineering approaches in microenvironment formation to let the authors better understand the big picture. Current Fig. 2 is too brief for non-experts.

3.      It will be useful to include/reuse some figures from representative studies for detailed discussion to showcase the current state of the art.

4.      The future direction section is relatively short. Please add more content and forward-looking thoughts/insights.  

5.      The authors self-cited 10 papers. I recommend including more works from others for diversity/inclusion.

Author Response

The review focuses on the important role of stem cell microenvironments in regenerative medicine, highlighting advancements in understanding and manipulating these environments for improved treatment strategies. It summarizes the role of the extracellular matrix, biochemical signals, and mechanical cues in influencing stem cell differentiation and function. In addition, this review also discusses innovative techniques such as 3D bioprinting and organ-on-a-chip models that mimic these complex microenvironments for enabling better understanding and control over tissue development and regeneration. The review also covers a bit of detail regarding the potential of these technologies in developing personalized medical treatments and their challenges, including ethical considerations and the need for further research to optimize stem cell-based therapies. Overall, this review underscores the importance of integrating multidisciplinary approaches to unlock the full potential of stem cell research in addressing current limitations in regenerative medicine. I think this review is worthy to be further considered. The authors shall better organize the content by tables/figures and include more relevant articles in the discussion. Please see my comments below.

  1. It will be helpful to include a glossary for footnotes to explain technical terms and abbreviations to make the content more accessible to non-specialists.

RE: We thank this suggestion and have included Table 1 that summarizes the most relevant technical terms and abbreviations used in the paper.

  1. Please provide a table explaining the pros/cons/limitations of different engineering approaches in microenvironment formation to let the authors better understand the big picture. Current Fig. 2 is too brief for non-experts.

RE: We thank this suggestion.

To address this issue, we have revised the figures.

Figure 1 was revised for quality and clarity.

We have included a new Figure 2 with schematics of the different engineering approaches for cell niche production in vitro. This figure highlights the fundamental aspects of each technique.

Figure 3 is now summarizing pros and cons of each technique mentioned in the text. We believe that this re-organization of figures obviates the need for a table.

  1. It will be useful to include/reuse some figures from representative studies for detailed discussion to showcase the current state of the art.

RE: This is a pertinent suggestion.

To address this, we have produced a new figure, Figure 4, which highlights a few examples referenced in the main text.

  1. The future direction section is relatively short. Please add more content and forward-looking thoughts/insights.

RE: We thank the reviewer for the comment, and we are aware that this section is shorter than other sections in the manuscript. However we believe that a longer section would affect the readability of the text.

Therefore, our intention was to create some curiosity in the reader about potential avenues for development in this area.

To do this, we have highlighted three potential points that we believe to be crucial in the future. These points are:

  • Improving bioink formulations for 3D bioprinting, in order to address current issues;
  • Developing and integrating new innovations in miniaturization of cell-based models, with the emphasizes on automated, high-throughput methods for studying cellular microenvironments and growth conditions in 3D;
  • Discussing the importance of finding ways to improve the reliability of in vitro assays in predicting in vivo responses, for example including sensing apparatus to the screening platforms.

This list represents our vision for the future and is of course subjective. Other points could be raised, but we believe that the section is reflecting our views for the field.

  1. The authors self-cited 10 papers. I recommend including more works from others for diversity/inclusion.

RE: We would like to thank the comment of the reviewer.

In fact, we self-cite 9 of our publications. However, in total, we refer to over 90 references. Thus, the vast majority of our citations is the work of others. We include our references to show our expertise in the field, but most of the review covers the work of other colleagues, at the expense of our own.

We therefore believe that we have a high degree of diversity and inclusion in our list of references.

Reviewer 3 Report

Comments and Suggestions for Authors

This review covers well-travelled territory but with an emphasis on biomechanical and tissue engineering topics as related to the stem cell microenvironment. It is well-organized and is expected to be of interest to the field.

Comments on the Quality of English Language

No significant issues 

Author Response

This review covers well-travelled territory but with an emphasis on biomechanical and tissue engineering topics as related to the stem cell microenvironment. It is well-organized and is expected to be of interest to the field.

RE: We would like to thank the positive remarks made by the reviewer. We have made additional revisions to further improve the manuscript.

Round 2

Reviewer 1 Report

Comments and Suggestions for Authors

It seems more acceptable now.

Reviewer 2 Report

Comments and Suggestions for Authors

The authors addressed most of my comments. The only concern is that the % of self-citation is still a bit high (~10%). I will leave it to the editor to decide if the MS can be accepted in the current form.